# Scalable Deep Generative Relational Models with High-Order Node Dependence

**Xuhui Fan**[1], **Bin Li**[2], **Scott A. Sisson**[1], **Caoyuan Li**[3], and **Ling Chen**[3]

[1]School of Mathematics & Statistics, University of New South Wales, Sydney
[2]Shanghai Key Lab of IIP & School of Computer Science, Fudan University
[3]Faculty of Engineering and IT, University of Technology, Sydney
*{xuhui.fan, scott.sisson}@unsw.edu.au; libin@fudan.edu.cn*

## Abstract

We propose a probabilistic framework for modelling and exploring the latent structure of relational data. Given feature information for the nodes in a network, the scalable deep generative relational model (SDREM) builds a deep network architecture that can approximate potential nonlinear mappings between nodes' feature information and the nodes' latent representations. Our contribution is two-fold: (1) We incorporate high-order neighbourhood structure information to generate the latent representations at each node, which vary smoothly over the network. (2) Due to the Dirichlet random variable structure of the latent representations, we introduce a novel data augmentation trick which permits efficient Gibbs sampling. The SDREM can be used for large sparse networks as its computational cost scales with the number of positive links. We demonstrate its competitive performance through improved link prediction performance on a range of real-world datasets.

## 1 Introduction

Bayesian relational models, which describe the pairwise interactions between nodes in a network, have gained tremendous attention in recent years, with numerous methods developed to model the complex dependencies within relational data; in particular, probabilistic Bayesian methods [27, 18, 1, 25, 7, 6]. Such models have been applied to community detection [27, 17], collaborative filtering [29, 23], knowledge graph completion [14] and protein-to-protein interactions [16]. In general, the goal of these Bayesian relational models is to discover the complex latent structure underlying the relational data and predict the unknown pairwise links [9, 8].

Despite improving the understanding of complex networks, existing models typically have one or more weaknesses: (1) While data commonly exhibit high-order node dependencies within the network, such dependencies are rarely modelled due to limited model capabilities; (2) Although a node's feature information closely informs its latent representation, existing models are not sufficiently flexible to describe these (potentially nonlinear) mappings well; (3) While some scalable network modelling techniques (e.g. Ber-Poisson link functions [30, 36]) can help to reduce the computational complexity to the number of positive links, they require the elements of latent representations to be independently generated and cannot be used for modelling dependent variables (e.g. membership distributions on communities).

In order to address these challenges, we develop a probabilistic framework using a deep network architecture on the nodes to model the relational data. The proposed scalable deep generative relational model (SDREM) builds a deep network architecture to efficiently map the nodes' feature information to their latent representations. In particular, the latent representations are modelled via Dirichlet

distributions, which permits their interpretation as membership distributions on communities. Based on the output latent representations (i.e. membership distributions) and an introduced community compatibility matrix, the relational data is modelled through the Ber-Poisson link function [30, 36], for which the computational cost scales with the number of positive links in the network.

We make two novel contributions: First, as the nodes' latent representations are Dirichlet random variables, we incorporate the full neighbourhood's structure information into its concentration parameters. In this way, high-order node dependence can be modelled well and can vary smoothly over the network. Second, we introduce a new data augmentation trick that enables efficient Gibbs sampling on the Ber-Poisson link function due to the Dirichlet random variable structure of the latent representations. The SDREM can be used to analyse large sparse networks and may also be directly applied to other notable models to improve their scalability (e.g. the mixed-membership stochastic blockmodel (MMSB) [1] and its variants [22, 13, 19]).

In comparison to existing approaches, the SDREM has several advantages. (1) *Modelling high-order node dependence*: Propagating information between nodes' connected neighbourhoods can improve information sharing and dependence modelling between nodes. Also, it can largely reduce computational costs in contrast to considering all the pairwise nodes' dependence, as well as avoid spurious or redundant information complications from unrelated nodes. Moreover, the non-linear real-value propagation in the deep network architecture can help to approximate the complex nonlinear mapping between the node's feature information and its latent representations. (2) *Scalable modelling on relational data*: Our novel data augmentation trick permits an efficient Gibbs sampling implementation, with computational costs scaling with the number of positive network links only. (3) *Meaningful layer-wise latent representation*: Since the nodes' latent representations are generated from Dirichlet distributions, they are naturally interpretable as the nodes' memberships over latent communities.

In our analyses on a range of real-world relational datasets, we demonstrate that the SDREM can achieve superior performance compared to traditional Bayesian methods for relational data, and perform competitively with other approaches. As the SDREM is the first Bayesian relational model to use neighbourhood-wise propagation to build the deep network architecture, we note that it may straightforwardly integrate other Bayesian methods for modelling high-order node dependencies in relational data, and further improve relationship predictability.

## 2  Scalable Deep Generative Relational Models (SDREMs)

The relational data in the SDREM is represented as a binary matrix $\boldsymbol{R} \in \{0, 1\}^{N \times N}$, where $N$ is the number of nodes and the element $R_{ij}$ ($\forall i, j$) indicates whether node $i$ relates to node $j$ ($R_{ij} = 1$ if the relation exists, otherwise $R_{ij} = 0$), with the self-connection relation $R_{ii}$ not considered here. The matrix $\boldsymbol{R}$ can be symmetric (i.e. undirected) or asymmetric (i.e. directed). The network's feature information is denoted by a non-negative matrix $\boldsymbol{F} \in \{\mathbb{R}^+ \cup 0\}^{N \times D}$, where $D$ denotes the number of features, and where each element $F_{id}$ ($\forall i, d$) takes the value of the $d$-th feature for the $i$-th node.

The deep network architecture of the SDREM is controlled by two parameters: $L$, representing the number of layers, and $K$, denoting the length of the nodes' latent representation in each layer. The latent representation $\boldsymbol{\pi}_i^{(l)}$ of node $i$ in the $l$-th layer is a Dirichlet random variable (i.e. a normalised vector with $(K-1)$ active elements). In this way, $\boldsymbol{\pi}_i^{(l)}$, which we term the "membership distribution", is interpretable as node $i$'s community distribution, where $K$ communities are modelled and $\pi_{ik}^{(l)}$ denotes node $i$'s interaction with the $k$-th community in the $l$-th layer.

The deep network architecture of the SDREM is composed of three parts: (1) The input layer feeding the feature information; (2) The hidden layers modelling high-order node dependences; (3) The output layer of the relational data model. These component parts are detailed below.

### 2.1  Feeding the feature information

When nodes' feature information is available, we introduce a feature-to-community transition coefficient matrix $\boldsymbol{T} \in (\mathbb{R}^+)^{D \times K}$, where $T_{dk}$ indicates the activity of the $d$-th feature in contributing to the $k$-th latent community. The linear sum of the transition coefficients $\boldsymbol{T}$ and feature $\boldsymbol{F}$ forms the

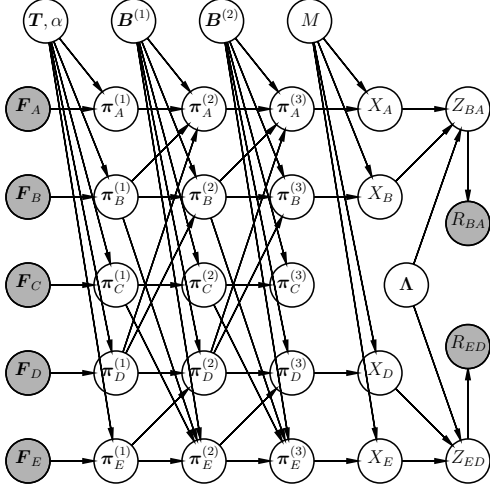

(1) $T_{dk} \sim \text{Gam}(\gamma_d^{(1)}, \frac{1}{c^{(1)}}), \boldsymbol{\pi}_i^{(1)} \sim \text{Dirichlet}(\boldsymbol{F}_i\boldsymbol{T} + \alpha);$

(2) For $l = 2, \dots, L$

$\bullet \quad B_{i'i}^{(l-1)} \begin{cases} \sim \text{Gam}(\gamma_1^{(l)}, \frac{1}{c^{(l)}}), & i' : R_{i'i} = 1; \\ \sim \text{Gam}(\gamma_0^{(l)}, \frac{1}{c^{(l)}}), & i' : i' = i; \\ = 0, & \text{otherwise}; \end{cases}$

$\bullet \quad \boldsymbol{\pi}_i^{(l)} \sim \text{Dirichlet}((\boldsymbol{B}_{\cdot i}^{(l-1)})^\top \cdot \boldsymbol{\pi}_{1:N}^{(l-1)}).$

(3) $M_i \sim \text{Poisson}(M), (X_{i1}, \dots, X_{iK}) \sim \text{Multi}(M_i; \pi_{i1}^{(L)}, \dots, \pi_{iK}^{(L)});$

(4) $\Lambda_{k_1 k_2} \sim \text{Gam}(k_\Lambda, \frac{1}{\theta_\Lambda});$

(5) $Z_{ij,k_1 k_2} \sim \text{Poisson}(X_{ik_1} \Lambda_{k_1 k_2} X_{jk_2});$

(6) $R_{ij} = \mathbf{1}(\sum_{k_1,k_2} Z_{ij,k_1 k_2} > 0).$

Figure 1: Illustration and visualization of a SDREM on a 5-node (i.e. $A, B, C, D, E$) directed network. Left: the graphical model of a 3-layer SDREM modelling $R_{BA}, R_{ED}$. Shaded nodes (i.e. $F_\cdot, R_\cdot$) denote variables with known values, unshaded nodes denote latent variables. Right top: the generative process of a SDREM. Right bottom: the directed connection types of all 5 nodes.

prior for the nodes' first layer membership distribution

$$T_{dk} \sim \text{Gam}(\gamma_d^{(1)}, \frac{1}{c^{(1)}}) \quad \forall d, k; \quad \boldsymbol{\pi}_i^{(1)} \sim \text{Dirichlet}(\boldsymbol{F}_i\boldsymbol{T} + \alpha) \quad \forall i. \tag{1}$$

where $\text{Gam}(\gamma, 1/c)$ denotes a gamma random variable with mean $\gamma/c$ and variance $\gamma/c^2$; $\{\gamma_d^{(1)}\}_d$ and $c^{(1)}$ are the hyper-parameters for generating $\{T_{dk}\}_{d,k}$. From Eq. (1), nodes with close feature information have similar prior knowledge and similar generated membership distributions. A supplementary contribution $\alpha$ is included in case that a node has no feature information available. For node $i$ without feature information, we have $\boldsymbol{\pi}_i^{(1)} \sim \text{Dirichlet}(\alpha \cdot \mathbf{1}^{1 \times K})$, which is a common setting in Bayesian relational data modelling.

## 2.2 Modelling high-order node dependence

High-order node dependence is modelled within the deep network architecture of the SDREM. In general, node $i$'s membership distribution $\boldsymbol{\pi}_i^{(l)}$ is conditioned on the membership distributions at the $(l-1)$-th layer via an information propagation matrix $\boldsymbol{B}^{(l-1)} \in \{\mathbb{R}^+ \cup 0\}^{N \times N}$:

$$B_{i'i}^{(l-1)} \begin{cases} \sim \text{Gam}(\gamma_1^{(l)}, \frac{1}{c^{(l)}}) & \text{if } R_{i'i} = 1; \\ \sim \text{Gam}(\gamma_0^{(l)}, \frac{1}{c^{(l)}}) & \text{if } i' = i; \\ = 0 & \text{otherwise}, \end{cases} \quad \boldsymbol{\pi}_i^{(l)} \sim \text{Dirichlet}((\boldsymbol{B}_{\cdot i}^{(l-1)})^\top \cdot \boldsymbol{\pi}_{1:N}^{(l-1)}), \tag{2}$$

Following [35], we set the hyper-parameter distribution as $\gamma_1^{(l)}, \gamma_0^{(l)} \sim \text{Gam}(e_0^{(l)}, \frac{1}{f_0^{(l)}}), c^{(l)} \sim \text{Gam}(g_0, \frac{1}{h_0})$. $B_{i'i}^{(l-1)}$ denotes node $i'$'s influence on node $i$ from the $(l-1)$-th to the $l$-th layer (e.g. larger values of $B_{i'i}^{(l-1)}$ will make $\boldsymbol{\pi}_i^{(l)}$ more similar to $\boldsymbol{\pi}_{i'}^{(l-1)}$) and $\boldsymbol{\pi}_{1:N}^{(l)} \in \{\mathbb{R}^+\}^{N \times K}$ denotes the matrix of $N$ nodes' membership distributions at the $l$-th layer. When there is no direct connection from node $i'$ to node $i$ (i.e. $i' \neq i \cap R_{i'i} = 0$), we restrict the corresponding information propagation coefficients $B_{i'i}$ at all layers to be 0; otherwise, we generate $B_{i'i}^{(l-1)}$ either from a node and layer-specified Gamma distribution (when $R_{i'i} = 1$) or a layer-specified Gamma distribution (when $i' = i$). This can produce various benefits. On one hand, it promotes the sparseness of $\boldsymbol{B}^{(l)}$ and reduces the cost of calculating $\boldsymbol{B}^{(l)}$ from $\mathcal{O}(N^2)$ to the scale of the number of positive network links. On the other hand, since the SDREM uses a Dirichlet distribution (parameterised by the linear

sum of node $i$'s neighbourhoods' membership distributions at the $(l-1)$-th layer) to generate $\boldsymbol{\pi}_i^{(l)}$, all the nodes' membership distributions are expected to vary smoothly over the connected graph structure. That is, connected nodes are expected to have more similar membership distributions than unconnected ones.

**Flexibility in modelling variance and covariance in membership distributions** Neighbourhood-wise information propagation allows for more flexible modelling than the extreme case of independent propagation whereby $\boldsymbol{\pi}_i^{(l)}$ is conditioned on $\boldsymbol{\pi}_i^{(l-1)}$ only (i.e. $\{\boldsymbol{B}^{(l)}\}_l$ is a diagonal matrix). Under independent propagation, the expected membership distribution at each layer does not change: $\mathbb{E}[\boldsymbol{\pi}_{1:N}^{(l)}] = \boldsymbol{\pi}_{1:N}^{(1)}$. In the SDREM, we have $\mathbb{E}[\boldsymbol{\pi}_{1:N}^{(l)}] = [\prod_{l'=1}^{l-1}(D^{(l')})^{-1}(\boldsymbol{B}^{(l')})^\top]\boldsymbol{\pi}_{1:N}^{(1)}$, where $D^{(l)}$ is a level $l$ diagonal matrix with $D_{ii}^{(l)} = \sum_{i'} B_{i'i}^{(l)}, \forall i$. Based on different choices for $\{\boldsymbol{B}^{(l)}\}_l$, the expected mean of each node's membership distribution can incorporate information from other nodes' input layer. In terms of variance and covariance within each $\boldsymbol{\pi}_i^{(l)}$, independent propagation is restricted to inducing a larger variance in $\pi_{ik}^{(l)}$ and smaller covariance between $\pi_{ik_1}^{(l)}$ and $\pi_{ik_2}^{(l)}$ due to the layer stacking architecture (this can be easily verified through the law of total variance and the law of total covariance). In contrast, for the SDREM, these variances and covariances can be made either large or small depending on the choices of $\{\boldsymbol{B}^{(l)}\}_l$ through the deep network architecture.

The Dirichlet distribution models the membership distribution $\{\boldsymbol{\pi}_i^{(l)}\}_{i,l}$ in a non-linear way. As non-linearities are easily captured via deep learning, it is expected that the deep network architecture in the SDREM can approximate the complex nonlinear mapping between the nodes' feature information and membership distributions sufficiently well. Further, the technique of propagating real-valued distributions through different layers might be a promising alternative to sigmoid belief networks [10, 11, 15], which mainly propagate binary variables between different layers.

**Comparison with spatial graph convolutional networks:** Propagating information through neighbourhoods works in a similar spirit to the spatial graph convolutional network (GCN) [2, 5, 12, 3] in a frequentist setting. In addition to providing variability estimates for all latent variables and predictions, the SDREM may conveniently incorporate beliefs on the parameters and exploit the rich structure within the data. Beyond the likelihood function, the SDREM uses a Dirichlet distribution as the activation function, whereas GCN algorithms usually use the logistic function. The resulting membership distribution representation of the SDREM may provide a more intuitive interpretation than the node representation (node embedding) in the GCN.

## 2.3 Scalable relational data modelling

We model the final-layer relational data via the Ber-Poisson link function [30, 36], $R_{ij} \sim$ Bernoulli$(1 - e^{-\sum_{k_1 k_2} X_{ik_1} \Lambda_{k_1 k_2} X_{jk_2}})$, where $X_{ik}$ is the latent count of node $i$ on community $k$ and $\Lambda_{k_1 k_2} \in \mathbb{R}^+$ is a compatibility value between communities $k_1$ and $k_2$. In existing work with the Ber-Poisson link function, all of the $\{X_{ik}\}_{i,k}$ terms are required to be independently generated (either from a Gamma [36, 34] or Bernoulli distribution [15]) to allow for efficient Gibbs sampling. However, in the SDREM, the elements of the output latent representation $(\pi_{i1}, \ldots, \pi_{iK})$ are jointly generated from a Dirichlet distribution. These normalised elements are dependent on each other and it is not easy to enable Gibbs sampling for each individual element $\{\pi_{ik}\}_k$.

To address this problem, we use a decomposition strategy to isolate the elements $\{\pi_{ik}\}_k$. We use multinomial distributions, with $\{\boldsymbol{\pi}_i\}_i$ as event probabilities, to generate $K$-length counting vectors $\{\boldsymbol{X}_i\}_i$. Each $\boldsymbol{X}_i$ can be regarded as an estimator of $\boldsymbol{\pi}_i$. Since the sum of the $\{X_{ik}\}_k$ is fixed as the number of trials (denoted as $M_i$) in the multinomial distribution, we further let $M_i$ be generated as $M_i \sim$ Poisson$(M)$. Based on the Poisson-Multinomial equivalence [4], each $X_{ik}$ is then equivalently distributed $X_{ik} \sim$ Poisson$(M\pi_{ik})$.

Following the settings of Ber-Poisson link function, a latent integer matrix $\boldsymbol{Z}_{ij} \in \mathbb{N}^{K \times K}$ is introduced, where the $(k_1, k_2)$-th entry is $Z_{ij,k_1 k_2} \sim$ Poisson$(X_{ik_1} \Lambda_{k_1 k_2} X_{jk_2})$. $R_{ij}$ is then generated by

evaluating the degree of positivity of the matrix $Z_{ij}$. That is, $\forall (i,j), k_1, k_2$:

$$M_i \sim \text{Poisson}(M), \quad (X_{i1}, \ldots, X_{iK}) \sim \text{Multi}(M_i; \pi_{i1}^{(L)}, \ldots, \pi_{iK}^{(L)}), \quad \Lambda_{k_1 k_2} \sim \text{Gam}(k_\Lambda, \frac{1}{\theta_\Lambda}),$$

$$Z_{ij,k_1 k_2} \sim \text{Poisson}(X_{ik_1} \Lambda_{k_1 k_2} X_{jk_2}) \quad \text{and} \quad R_{ij} = \mathbf{1}(\sum_{k_1,k_2} Z_{ij,k_1 k_2} > 0). \tag{3}$$

Here, the prior distribution for generating $X_{ik}$ and the likelihood based on $X_{ik}$ are both Poisson distributions. Consequently, we may implement posterior sampling by using Touchard polynomials [31] (details in Section 3).

To model binary or count data, the Ber-Poisson link function [30, 36] decomposes the latent counting vector $\boldsymbol{X}_i$ into the latent integer matrix $\boldsymbol{Z}_{ij}$. An appealing property of this construction is that we do not need to calculate the latent integers $\{z_{ij,k_1 k_2}\}_{k_1,k_2}$ over the 0-valued $R_{ij}$ data as they are equal to 0 almost surely. Hence, the focus can be on the positive-valued relational data. This is particularly useful for real-world network data as usually only a small fraction of the data is positive. Hence, the computational cost for inference scales only with the number of positive relational links.

When nodes' feature information is not available (i.e. $\boldsymbol{F} = 0^{N \times D}$) and $L = 1$, the SDREM reduces to the same settings as the MMSB [1]. In particular, the membership distributions of both the MMSB and the SDREM follow the same Dirichlet distribution $\{\boldsymbol{\pi}_i\}_i \sim \text{Dirichlet}(\alpha^{1 \times K})$. As the MMSB and its variants [22, 13, 19] introduce pairwise latent labels for all the relational data (both 1 and 0-valued data), it requires a computational cost of $\mathcal{O}(N^2)$ to infer all latent variables. In contrast, our novel data augmentation trick can be straightforwardly applied in these models (by simply replacing the Ber-Beta likelihood [27, 18] with Ber-Poisson link function) and reduce their computational cost to the scale of the number of positive links. We show in Section 5 that we can also get better predictive performance with this strategy.

## 3 Inference

The joint distribution of the relational data and all latent variables in the SDREM is:

$$P(\{\boldsymbol{\pi}_i^{(l)}\}_{i,l}, \{\boldsymbol{B}^{(l)}\}_l, \boldsymbol{\Lambda}, \{Z_{ij,k_1 k_2}\}_{i,j,k_1,k_2}, \{R_{ij}\}_{i,j}, \{X_{ik}\}_{i,k}, \boldsymbol{T} | \boldsymbol{F}, \boldsymbol{\gamma}, \boldsymbol{c}, \alpha, M, k_\Lambda, \theta_\Lambda)$$

$$= \left[ \prod_{i=1}^n P(\boldsymbol{\pi}_i^{(1)} | \alpha, \boldsymbol{F}_i, \boldsymbol{T}) \right] \prod_{l=1}^{L-1} \left[ P(\boldsymbol{B}^{(l)} | \gamma_i^{(l)}, c^{(l)}) \prod_{i=1}^n P(\boldsymbol{\pi}_i^{(l+1)} | \{\boldsymbol{\pi}_{i'}^{(l)}\}_{i':R_{i'i}=1}, \boldsymbol{\pi}_i^{(l)}, \boldsymbol{B}^{(l)}) \right] P(\boldsymbol{\Lambda} | k_\Lambda, \theta_\Lambda)$$

$$\times \left[ \prod_{i,k} P(X_{ik} | \pi_{ik}^{(L)}, M) \right] \left[ \prod_{(i,j)|R_{ij}=1, k_1, k_2} P(Z_{ij,k_1 k_2} | X_{ik_1}, X_{jk_2}, \Lambda_{k_1 k_2}) \right] \left[ \prod_{f,k} P(T_{dk} | \gamma_f^{(1)}, c^{(1)}) \right]. \tag{4}$$

By introducing auxiliary variables, all latent variables can be sampled via efficient Gibbs sampling. This section focuses on inference for $\{X_{ik}\}_{i,k}$, which is the key variable involving the data augmentation trick. Sampling the membership distributions $\{\boldsymbol{\pi}_i^{(l)}\}_{i,l}$ is as implemented in Gamma Belief Networks [37] and Dirichlet Belief Networks [35], which mainly use a bottom-up mechanism to propagate the latent count information in each layer. As sampling the other variables is trivial, we relegate the full sampling scheme to the Supplementary Material (Appendix A).

**Sampling $\{X_{ik}\}_{i,k}$:** From the Poisson-Multinomial equivalence [4] we have $M_i \sim \text{Poisson}(M)$,

$$(X_{i1}, \ldots, X_{iK}) \sim \text{Multi}(M_i; \pi_{i1}^{(L)}, \ldots, \pi_{iK}^{(L)}) \stackrel{d}{=} X_{ik} \sim \text{Poisson}(M\pi_{ik}^{(L)}), \forall k.$$

Both the prior distribution for generating $X_{ik}$ and the likelihood parametrised by $X_{ik}$ are Poisson distributions. The full conditional distribution of $X_{ik}$ (assuming $z_{ii,..} = 0, \forall i$) is then

$$P(X_{ik} | M, \boldsymbol{\pi}, \boldsymbol{\Lambda}, \boldsymbol{Z}) \propto \frac{\left[ M\pi_{ik}^{(L)} e^{-\sum_{j \neq i, k_2} X_{jk_2}(\Lambda_{kk_2} + \Lambda_{k_2 k})} \right]^{X_{ik}}}{X_{ik}!} (X_{ik})^{\sum_{j_1, k_2} Z_{ij_1, kk_2} + \sum_{j_2, k_1} Z_{j_2 i, k_1 k}}. \tag{5}$$

This follows the form of Touchard polynomials [31], where $1 = \frac{1}{e^x T_n(x)} \sum_{k=0}^{\infty} \frac{x^k k^n}{k!}$ with $T_n(x) = \sum_{k=0}^n \{{n \atop k}\} x^k$ and where $\{{n \atop k}\}$ is the Stirling number of the second kind. A draw from (5) is then available by comparing a Uniform$(0, 1)$ random variable to the cumulative sum of $\{\frac{1}{e^x T_n(x)} \cdot \frac{x^k k^n}{k!}\}_k$.

# 4 Related Work

There is a long history of using Bayesian methods for relational data. Usually, these models build latent representations for the nodes and use the interactions between these representations to model the relational data. Typical examples include the stochastic blockmodel [27, 26, 18] (which uses latent labels), the mixed-membership stochastic blockmodel (MMSB) [1, 22] (which uses membership distributions) and the latent feature relational model (LFRM) [25, 28] (which uses binary latent features). As most of these approaches are constructed using shallow models, their modelling capability is limited.

The Multiscale-MMSB [13] is a related model, which uses a nested-Chinese Restaurant Process to construct hierarchical community structures. However, its tree-type structure is quite complicated and hard to implement efficiently. The Nonparametric Metadata Dependent Relational model (NMDR) [19] and the Node Attribute Relational Model (NARM) [34] also use the idea of transforming nodes' feature information to nodes' latent representations. However, because of their shallow latent representation, these methods are unable to describe higher-order node dependencies.

The hierarchical latent feature model (HLFM) [15] may be the closest model to the SDREM, as they each build up deep network architecture to model relational data. However, the HLFM uses a sigmoid belief network, and does not consider high-order node dependencies, so that each node only depends on itself through layers. Finally, feature information enters in the last layer of the deep network architecture, and so the HLFM is unable to sufficiently describe nonlinear mappings between the feature information and the latent representation.

Recent developments [10, 11] in Poisson matrix factorisation also try to build deep network architecture for latent structure modelling. Since these mainly use sigmoid belief networks, the way of propagating binary variables is different from our real-valued distributions propagation. Information propagation through Dirichlet distributions in the SDREM follows the approaches of [37][35]. However, their focus is on topic modelling and no neighbourhood-wise propagation is discussed in these methods.

Our SDREM shares similar spirit of the Variational Graph Auto-Encoder (VGAE) [21, 24] algorithms. Both of the algorithms aim at combining the graph convolutional networks with Bayesian relational methods. However, VGAE has a larger computational complexity ($\mathcal{O}(N^2)$). It uses parameterized functions to construct the deep network architecture and the probabilistic nature occurs in the output layer as Gaussian random variables only. In contrast, SDREM constructs multi-stochastic-layer architectures (with Dirichlet random variables at each layer). Thus, SDREM would have better model interpretations (see Figure 5).

We note that recent work [33] also claims to estimate uncertainty in the graph convolutional neural networks setting. This work uses a two-stage strategy: it firstly takes the observed network as a realisation from a parametric Bayesian relational model, and then uses Bayesian Neural Networks to infer the model parameters. The final result is a posterior distribution over these variables. Unlike the SDREM, this work performs the inference in two stages and also lacks inferential interpretability.

**Computational complexities**   The computational complexity of the SDREM is $\mathcal{O}(NDK+(NK+N_E)L + N_E K^2)$ and scales to the number of positive links, $N_E$. In particular, $\mathcal{O}(NDK)$ refers to the feature information incorporation in the input layer, $\mathcal{O}((NK + N_E)L)$ refers to the information propagation in the deep network architecture and $\mathcal{O}(N_E K^2)$ refers to the relational data modelling in the output layer. The SDREM's computational complexity is comparable to that of the HLFM, which is $\mathcal{O}(NDK + NKL + N_E K^2)$, and the NARM, which is $\mathcal{O}(NDK + N_E K^2)$ [34] and is significantly less than that of the MMSB-type algorithms.

# 5 Experiments

**Dataset Information**   In the following, we examine four real-world datasets: three standard citation networks (*Citeer*, *Cora*, *Pubmed* [32] and one protein-to-protein interaction network (*PPI*) [38]. Summary statistics for these datasets are displayed in Table 1. In the citation datasets, nodes correspond to documents and edges represent citation links. A node's features comprise the documents' bag-of-words representations. In the protein-to-protein dataset, we use the pre-processed feature information provided by [12].

Table 1: Dataset information. $N$ is the number of nodes, $N_E$ is the number of positive links, $D$ is the number of features, F.D.= # nonzeros entries/# total entries in $F$ and it refers to the density of features.

| Dataset | $N$ | $N_E$ | $D$ | F.D. | Dataset | $N$ | $N_E$ | $D$ | F.D. |
|---------|-----|-------|-----|------|---------|-----|-------|-----|------|
| Citeer | $3,312$ | $4,715$ | $3,703$ | $0.86\%$ | Cora | $2,708$ | $5,429$ | $1,433$ | $1.27\%$ |
| Pubmed | $2,000$ | $17,522$ | $500$ | $1.80\%$ | PPI | $4,000$ | $105,775$ | $50$ | $10.20\%$ |

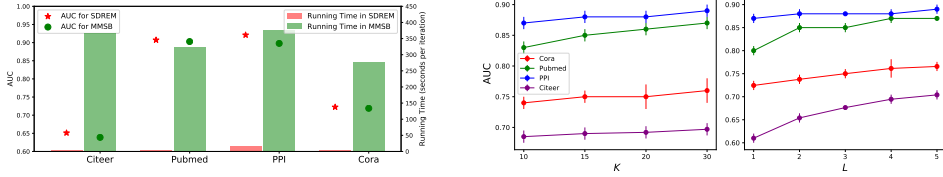

Figure 2: Left: Mean AUC (dots) and per iteration computing time (bar heights) comparison between the simplified SDREM and the MMSB for each dataset. Right: Mean AUC performance as a function of the number of membership distributions ($K$; with $L = 3$) and the number of layers ($L$; with $K = 20$).

**Evaluation Criteria**  We primarily focus on link prediction and use this to evaluate model performance. We use AUC (Area Under ROC Curve) and Average Negative-Log-likelihood on test relational data as the two comparison criteria. The AUC value represents the probability that the algorithm will rank a randomly chosen existing-link higher than a randomly chosen non-existing link. Therefore, the higher the AUC value, the better the predictive performance. For hyper-parameters we specify $M \sim \text{Gam}(N, 1)$ for all datasets, and $\{\gamma_d^{(1)}\}_d, \{\gamma_1^{(l)}, \gamma_0^{(l)}\}_l, \{c^{(l)}\}_l$ are all given $\text{Gam}(1, 1)$ priors. Each reported criteria value is the mean of 10 replicate analyses. Each replicate uses 2000 MCMC iterations with the first 1000 discarded as burn-in. Unless specified, reported AUC values are obtained by using $90\%$ (per row) of the data as training data and the remaining $10\%$ as test data. The testing relational data are not used when constructing the information propagation matrix (i.e. we set $\{\beta_{i'i}^{(l)}\}_l = 0$ if $R_{i'i}$ is testing data).

**Validating the data augmentation trick:**  We first evaluate the effectiveness of the data augmentation trick through comparisons with the MMSB [1]. To make a fair comparison, we specify the SDREM as $\boldsymbol{F} = 0^{N \times 1}, L = 1, K = 20$, so that the membership distributions in each model follow the same Dirichlet distribution $\{\boldsymbol{\pi}_i\}_i \sim \text{Dirichlet}(\alpha \cdot \mathbf{1}^{1 \times 20})$. Figure 2 (left panel) displays the mean AUC and per iteration running time for these two models. It is clear that the AUC values of the simplified SDREM are always better than those of the MMSB, and the time required for one iteration in the SDREM is substantially lower (at least two orders of magnitude lower) than that of the MMSB. Note that the running time of the SDREM is highest for the PPI dataset, since it contains the largest number of positive links and the computational cost of the SDREM scales with this value.

**Different settings of $K$ and $L$:**  We evaluate the SDREM's behaviour under different architecture settings, through the influence of two parameters: $K$, the length of the membership distributions, and $L$, the number of layers. When testing the effect of different values of $K$ we fixed $L = 3$, and when varying $L$ we fixed $K = 20$. Figure 2 (right panel) displays the resulting mean AUC values under these settings. As might be expected, the SDREM's AUC value increases with higher model complexity (i.e. larger values of $K$ and $L$). The worst performance occurs with $L = 1$ layer as it has the least flexible modelling capability. Considering the computational complexity and modelling power, we set $K = 20$ and $L = 4$ for the remaining analyses in this paper.

**Deep network architecture:**  We evaluate the advantage of using neighbourhood connections to propagate layer-wise information. Three different deep network architectures are compared: (1) *Plain-SDREM*. We assume the nodes' feature information is unavailable and use an identity matrix to represent the features (i.e. $\boldsymbol{F} = I_{N \times N}$) (we tried two cases, $\boldsymbol{F} = 0^{N \times 1}$ and $\boldsymbol{F} = I_{N \times N}$ and found the latter to perform better). (2) *Fully-connected-SDREM (Full-SDREM)*. The propagation coefficient $B_{i'i}^{(l)}$ is not restricted to be 0 when $R_{i'i} = 0$ and instead a hierarchical Gamma process is specified as a sparse prior on all the propagation coefficients. (3) *Independent-SDREM (Inde-SDREM)*.

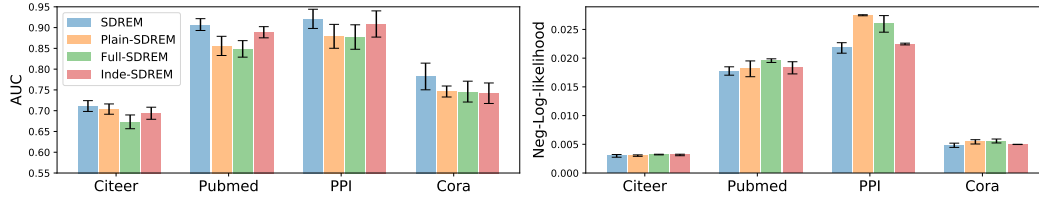

Figure 3: Mean AUC ($\pm 1.96\times$ standard errors (of the mean)) and negative Log-Likelihood ($\pm 1.96\times$ standard errors) on 10% test data for each dataset.

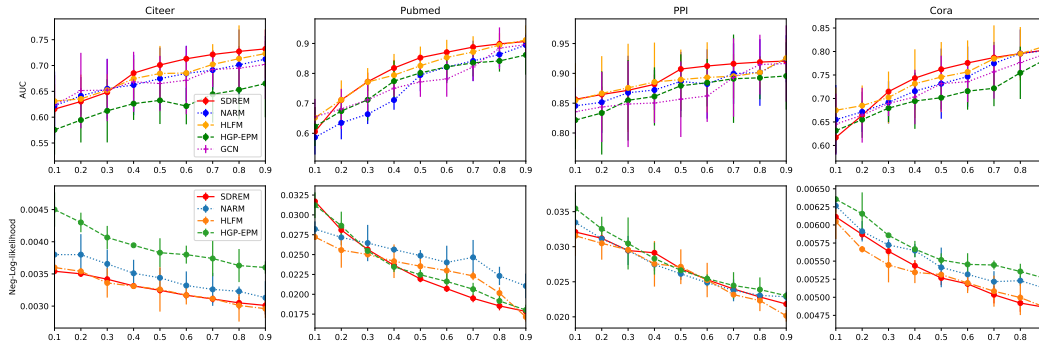

Figure 4: Mean AUC and negative Log-Likelihood values (points) as a function of the proportion of training data ($x$-axis), for each dataset and deep network architecture. Vertical lines correspond to the 95% confidence interval of reported statistics $\pm 1.96\times$ standard error.

This assumes each node propagates information only to itself and does not exchange information with other nodes in the deep network architecture (i.e. each $\{\boldsymbol{B}^{(l)}\}_l$ is a diagonal matrix).

Figure 3 shows the performance of each of these different configurations against the non-restricted SDREM. It is clear that the non-restricted SDREM achieves the best performance in both mean AUC and negative-Log-Likelihood among all network configurations. The Full-SDREM consistently performs the worst among all configurations. This suggests that the fully connected architecture is a poor candidate, and the sampler may become easily be trapped in local modes.

**Performance in the presence of feature information:** We compare the SDREM with several alternative Bayesian methods for relational data and one Graph Convolutional Network model. We examine: the Hierarchical Latent Feature Relational Model (HLFM) [15], the Node Attribute Relational Model (NARM) [34], the Hierarchical Gamma Process-Edge Partition Model (HGP-EPM) [36] and a graph convolutional neural network (GCN) [20]. The NARM, HGP-EPM and GCN methods are executed using their respective authors' implementations, under their default settings. The HLFM is implemented to the best of our abilities and we set the same number of layers and length of latent binary representation as the SDREM. For the GCN, the AUC value is calculated based on the pairwise similarities between the node representations and the ground-truth relational data and the Negative Log-Likelihood is unavailable due to its frequentist setting.

Figure 4 shows the performance of each method on the four datasets, under different ratios of training data ($x$-axis). In terms of AUC, the SDREM performs the best among all the methods when the proportion of training data ratio is larger than $0.5$. However, the performance of the SDREM is not outstanding when the training data ratio is less than $0.5$. This may partly be due to there being insufficient relational data to effectively model the latent counts. Since the SDREM and the HLFM are the best performing two algorithms in most cases, this confirms the effectiveness of utilising a deep network architecture. Similarly conclusions can be drawn based on the negative log-likelihood: the SDREM and the HLFM are the best performing two algorithms.

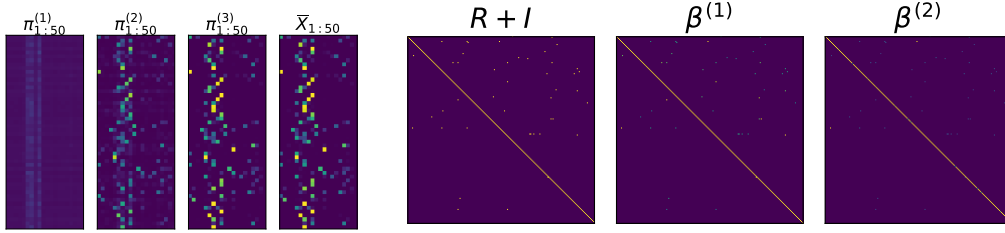

Figure 5: Left: visualizations on the membership distributions ($\{\boldsymbol{\pi}_{1:50}^{(l)}\}_{l=1}^{3}$) and normalized auxiliary counting variable ($\bar{\boldsymbol{X}}_{1:50}$) for the first 50 nodes of the *Citeer* dataset (row represents the nodes and column represents the latent features); right: visualizations on the non-zero positions ($\boldsymbol{R} + \boldsymbol{I}$) and transition coefficient matrix ($\{\boldsymbol{\beta}^{(l)}\}_{l=1}^{2}$) for the first 200 nodes of the *Citeer* dataset.

Table 2: Average latent counts (per node) in different layers.

| Dataset | Layer 3 | Layer 2 | Layer 1 | Dataset | Layer 3 | Layer 2 | Layer 1 |
|---------|---------|---------|---------|---------|---------|---------|---------|
| Citeer | 533.7 | 7.8 | 2.5 | Cora | 290.1 | 7.0 | 2.3 |
| Pubmed | 292.4 | 24.8 | 10.1 | PPI | 65.6 | 20.1 | 12.7 |

**Comparison with Variational Graph Auto-Encoder** We also make brief comparisons with the Variational Graph Auto-Encoder (VGAE) [21]. Taking $90\%$ of the data as training data and the remaining as testing data, the average AUC scores of 16 random VGAE runs for these datasets are: *Citeseer* (0.863), *Cora* (0.854), *Pubmed* (0.921) and *PPI* (0.934). Considering the attributes of these datasets, we find that VGAE obtains a better performance than our SDREM in the datasets with sparse linkages, whereas their performance in other types of datasets are competitive. This phenomenon might be caused by two reasons: (1) due to the inference nature (backward latent counts propagating and forward variable sampling), our SDREM propagates less counting information (see Table 2) to higher layers. The deep hierarchical structure might be less powerful in sparse networks; (2) the Sigmod and ReLu activation functions might be more flexible than the Dirichlet distribution for the case of sparse networks. We will keep on investigating this issue in the future work.

**Latent structure visualization:** We also visualize the latent structures of the model to get further insights in Figure 5. According to the left panel, we can see that the membership distributions gradually become more distinguished along with the layers. The less distinguished membership distributions might due to two reasons: (1) the higher abstraction of the latent features; (2) the insufficient latent counting information back-propagated to these higher layers. The normalized latent counting vector ($\boldsymbol{X}$) looks to be identical to the output membership distribution $\boldsymbol{\pi}^{(3)}$. This verifies that our introduction of $\boldsymbol{X}$ seems to successfully pass the information to the latent integers variable $\boldsymbol{Z}$. In the right panel of information propagation matrix, we can see that the neighbourhood-wise information seems to become weaker from the input layer to the output layer.

## 6 Conclusion

We have introduced a Bayesian framework by using deep latent representations for nodes to model relational data. Through efficient neighbourhood-wise information propagation in the deep network architecture and a novel data augmentation trick, the proposed SDREM is a promising approach for modelling scalable networks. As the SDREM can provide variability estimates for its latent variables and predictions, it has the potential to be a competitive alternative to frequentist graph convolutional network-type algorithms. The promising experimental results validate the effectiveness of the SDREM's deep network architecture and its competitive performance against other approaches. Since the SDREM is the first work to use neighbourhood-wise information propagation in Bayesian methods, combining this with other Bayesian relational models and other applications with pairwise data (e.g. collaborative filtering) would be interesting future work.

## Acknowledgements

Xuhui Fan and Scott A. Sisson are supported by the Australian Research Council through the Australian Centre of Excellence in Mathematical and Statistical Frontiers (ACEMS, CE140100049), and Scott A. Sisson through the Discovery Project Scheme (DP160102544). Bin Li is supported by Shanghai Municipal Science & Technology Commission (16JC1420401) and the Program for Professor of Special Appointment (Eastern Scholar) at Shanghai Institutions of Higher Learning.

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
