[Supplementary Material · SDREM_supp.pdf]

# Supplementary Material:
# Scalable Deep Generative Relational Models with High-Order Node Dependence

Xuhui Fan[1], Bin Li[2], Scott A. Sisson[1], Caoyuan Li[3], and Ling Chen[3]

[1]School of Mathematics & Statistics, University of New South Wales, Sydney
[2]Shanghai Key Lab of IIP & School of Computer Science, Fudan University
[3]Faculty of Engineering and IT, University of Technology, Sydney
*{xuhui.fan, scott.sisson}@unsw.edu.au; libin@fudan.edu.cn*

## 1 Inference algorithm

### 1.1 Back-propagate the hidden counts from the output layer to the input layer

We first back-propagate the hidden counts from the output layer to the input layer sequentially. In the output layer, $\{\boldsymbol{X}_i\}_i$ are regarded as the hidden counts and we denote as $\boldsymbol{X}_i = \boldsymbol{m}_i^{(L)}, \forall i$. For layer $l$ ($l = 1, \ldots, L-1$), we may first integrate $\boldsymbol{\pi}_i^{(l)}$ and obtain the likelihood term of the latent counts $\boldsymbol{m}_i^{(l)}$ for $\boldsymbol{\psi}_i^{(l)}$ ($\boldsymbol{\psi}_i^{(l)} = \sum_{i'i} B_{i'i}^{(l-1)}\boldsymbol{\pi}_{i'}^{(l-1)}$, ) as:

$$
\begin{aligned}
P(\{m_{ik}^{(l)}\}_k | \{\psi_{ik}^{(l)}\}_k) &= \int_{\boldsymbol{\pi}_i^{(l)}} P(\boldsymbol{m}_i^{(l)}|\boldsymbol{\pi}_i^{(l)}) P(\boldsymbol{\pi}_i^{(l)}|\boldsymbol{\psi}_i^{(l)}) d\boldsymbol{\pi}_i^{(l)} \\
&\propto \int_{\boldsymbol{\pi}_i^{(l)}} \prod_k [\pi_{ik}^{(l)}]^{m_{ik}^{(l)}} \cdot \frac{\Gamma(\sum_k \psi_{ik}^{(l)})}{\prod_k \Gamma(\psi_{ik}^{(l)})} \cdot [\pi_{ik}^{(l)}]^{\psi_{ik}^{(l)}-1} d\boldsymbol{\pi}_i^{(l)} \\
&= \frac{\Gamma(\sum_k \psi_{ik}^{(l)})}{\prod_k \Gamma(\psi_{ik}^{(l)})} \cdot \int_{\boldsymbol{\pi}_i^{(l)}} \prod_k [\pi_{ik}^{(l)}]^{m_{ik}^{(l)}+\psi_{ik}^{(l)}-1} d\boldsymbol{\pi}_i^{(l)} \\
&= \frac{\Gamma(\sum_k \psi_{ik}^{(l)})}{\Gamma(\sum_k \psi_{ik}^{(l)} + \sum_k m_{ik}^{(l)})} \cdot \prod_k \frac{\Gamma(\psi_{ik}^{(l)} + m_{ik}^{(l)})}{\Gamma(\psi_{ik}^{(l)})}
\end{aligned}
\tag{1}
$$

where $\Gamma(\cdot)$ is a Gamma function, $m_{ik}^{(l)}$ refers to the "hidden counts" for the $l$-th layer.

We introduce two types of auxiliary variables $q_i^{(l)}, \{y_{ik}^{(l)}\}_k$ and define the joint probability of $q_i^{(l)}, \{y_{ik}^{(l)}\}_k, \{m_{ik}^{(l)}\}_k$ as:

$$
\begin{aligned}
&P(q_i^{(l)}, \{y_{ik}^{(l)}\}_k, \{m_{ik}^{(l)}\}_k | \{\psi_{ik}^{(l)}\}_k) \\
=& \Gamma(\sum_k m_{ik}^{(l)}) \cdot [q_i^{(l)}]^{\sum_k \psi_{ik}^{(l)}-1} [1-q_i^{(l)}]^{\sum_k m_{ik}^{(l)}-1} \prod_k \left[ (\psi_{ik}^{(l)})^{y_{ik}^{(l)}} \cdot (m_{ik}^{(l)} - y_{ik}^{(l)})! \right]
\end{aligned}
\tag{2}
$$

where $q_i^{(l)} \in [0,1], y_{ik}^{(l)} \in \{1, \ldots, m_{ik}^{(l)}\}_k, m_{ik} \in \mathbb{N}^+$.

Integrating $q_i^{(l)}, \{y_{ik}^{(l)}\}_k$ in Eq. (2), we can see the marginal probability to $\{m_{ik}^{(l)}\}_k$ is $P(\{m_{ik}^{(l)}\}_k|\{\psi_{ik}^{(l)}\}_k)$.

$$\sum_{y_{i1}^{(l)}} \cdots \sum_{y_{iK}^{(l)}} \int_{q_i^{(l)}} P(q_i^{(l)}, \{y_{ik}^{(l)}\}_k, \{m_{ik}^{(l)}\}_k|\{\psi_{ik}^{(l)}\}_k) dq_i^{(l)} = P(\{m_{ik}^{(l)}\}_k|\{\psi_{ik}^{(l)}\}_k) \tag{3}$$

Since $\sum_{m_{i1}^{(l)}} \cdots \sum_{m_{iK}^{(l)}} P(\{m_{ik}^{(l)}\}_k|\{\psi_{ik}^{(l)}\}_k) = 1$, $P(q_i^{(l)}, \{y_{ik}^{(l)}\}_k, \{m_{ik}^{(l)}\}_k|\{\psi_{ik}^{(l)}\}_k)$ is a valid probability measure.

The individual conditional probabilities of $q_i^{(l)}$ and $y_{ik}^{(l)}$ in Eq. (2) are:

$$P(q_i^{(l)}|\{m_{ik}^{(l)}\}_k, \{\psi_{ik}^{(l)}\}_k) \propto [q_i^{(l)}]^{\sum_k \psi_{ik}^{(l)}-1}[1-q_i^{(l)}]^{\sum_k m_{ik}^{(l)}-1} \rightarrow q_i^{(l)} \sim \text{Beta}(\sum_k \psi_{ik}^{(l)}, \sum_k m_{ik}^{(l)})$$

$$P(y_{ik}^{(l)}|m_{ik}^{(l)}, \psi_{ik}^{(l)}) \propto (\psi_{ik}^{(l)})^{y_{ik}^{(l)}} \cdot (m_{ik}^{(l)} - y_{ik}^{(l)})! \rightarrow y_{ik}^{(l)} \sim \text{CRT}(m_{ik}^{(l)}, \psi_{ik}^{(l)}) \tag{4}$$

We abstract ther terms associated with $\{\psi_{ik}^{(l)}\}_k$ in Eq. (2) and we obtain the associated likelihood term as:

$$\mathcal{L}(\{\psi_{ik}^{(l)}\}_k|q_i^{(l)}, \{y_{ik}^{(l)}\}_k) \propto [q_i^{(l)}]^{\sum_k \psi_{ik}^{(l)}} \prod_k \left[ (\psi_{ik}^{(l)})^{y_{ik}^{(l)}} \right] \tag{5}$$

Based on the fact that $\psi_{ik}^{(l)} = \sum_{i'} \pi_{i'k}^{(l-1)} B_{i'i}^{(l-1)}$, we first decompose the integer $y_{ik}^{(l)}$ on $\psi_{ik}^{(l)}$ to the previous $(l-1)$-th layer as:

$$\left( h_{1ik}^{(l)}, \ldots, h_{Nik}^{(l)} \right) \sim \text{Multi} \left( y_{ik}^{(l)}; \frac{\pi_{1k}^{(l-1)} B_{1i}^{(l-1)}}{\psi_{ik}^{(l)}}, \ldots, \frac{\pi_{Nk}^{(l-1)} B_{Ni}^{(l-1)}}{\psi_{ik}^{(l)}} \right) \tag{6}$$

Combining Eq. (5)(6) and the fact that $\psi_{ik}^{(l)} = \sum_{i'} \pi_{i'k}^{(l-1)} B_{i'i}^{(l-1)}$, we obtain the associated likelihood term as:

$$P(\{h_{1ik}^{(l)}, \ldots, h_{Nik}^{(l)}\}_k, q_i^{(l)}|\{\pi_{i'k}^{(l-1)}\}_k, B_{i'i}^{(l-1)})$$

$$\propto [q_i^{(l)}]^{\sum_k \sum_{i'} \pi_{i'k}^{(l-1)} B_{i'i}^{(l-1)}} \prod_k \left[ (\sum_{i'} \pi_{i'k}^{(l-1)} B_{i'i}^{(l-1)})^{y_{ik}^{(l)}} \cdot \frac{\prod_{i'} (\pi_{i'k}^{(l-1)} B_{i'i}^{(l-1)})^{h_{i'ik}^{(l)}}}{(\sum_{i'} \pi_{i'k}^{(l-1)} B_{i'i}^{(l-1)})^{y_{ik}^{(l)}}} \right]$$

$$= [q_i^{(l)}]^{\sum_{i'} B_{i'i}^{(l-1)}} \prod_k \prod_{i'} (\pi_{i'k}^{(l-1)} B_{i'i}^{(l-1)})^{h_{i'ik}^{(l)}}$$

$$= [q_i^{(l)}]^{\sum_{i'} B_{i'i}^{(l-1)}} \prod_{i'} (B_{i'i}^{(l-1)})^{\sum_k h_{i'ik}^{(l)}} \cdot \prod_k \prod_{i'} (\pi_{i'k}^{(l-1)})^{h_{i'ik}^{(l)}} \tag{7}$$

The conditional probability given $\{\pi_{i'k}^{(l-1)}\}_k$ is unrelated to $q_i^{(l)}$ and $B_{i'i}^{(l)}$. We can simplify it as:

$$P(\{h_{1ik}^{(l)}, \ldots, h_{Nik}^{(l)}\}_k|\{\pi_{i'k}^{(l-1)}\}_k) \propto \prod_{i'} \left[ \prod_k (\pi_{i'k}^{(l-1)})^{h_{i'ik}^{(l)}} \right] \tag{8}$$

That is, $(h_{1ik}^{(l)}, \ldots, h_{Nik}^{(l)})$ can be regarded as the latent counts (i.e. works as a draw from a Multinomial distribution with $\boldsymbol{\pi}_{i'}^{(l-1)}$ as event probabilities) and their summation $\{y_{ik}^{(l)}\}$ can be regarded as the latent count vector from the input count vector $\{m_{ik}^{(l)}\}$.

As a result, the latent count of the $(l-1)$-th layer can be summarized as

$$m_{i'k}^{(l-1)} = \sum_i h_{i'ik}^{(l)} \tag{9}$$

to represent the "hidden counts" in the $(l-1)$-th layer.

## 1.2 Posterior sampling in a top-down manner

**Sampling** $\{T_{dk}\}_{d,k}$    Using similar procedure, we first calculate $\psi_{ik}^{(1)} = \sum_d F_{id} T_{dk} + \alpha$ and then generate the variables of $q_i^{(1)}, \{y_{ik}^{(1)}\}_k$ as:

$$q_i^{(1)} \sim \text{Beta}(\sum_k \psi_{ik}^{(1)}, \sum_k m_{ik}^{(1)}), y_{ik}^{(1)} \sim \text{CRT}(m_{ik}^{(1)}, \psi_{ik}^{(1)}) \tag{10}$$

The latent count $y_{ik}^{(1)}$ on $\psi_{ik}^{(1)}$ to the input feature layer can be defined as:

$$\left(h_{i1k}^{(1)}, \ldots, h_{iDk}^{(1)}, h_{i\alpha k}^{(1)}\right) \sim \text{Multi}\left(y_{ik}^{(1)}; \frac{F_{i1} T_{1k}}{\psi_{ik}^{(1)}}, \ldots, \frac{F_{iD} T_{Dk}}{\psi_{ik}^{(1)}}, \frac{\alpha}{\psi_{ik}^{(1)}}\right) \tag{11}$$

With the associated likelihood (with regards to $T_{dk}$) displayed as:

$$\mathcal{L}(T_{dk}) \propto \left[\prod_i (q_i^{(1)})^{F_{id}}\right]^{T_{dk}} (T_{dk})^{\sum_i h_{idk}^{(1)}} \tag{12}$$

We get $T_{dk}$'s posterior distribution as

$$T_{dk} \sim \text{Gam}(k_T + \sum_i h_{idk}^{(1)}, \frac{1}{\theta_T - \sum_i F_{id} \log q_i^{(1)}}) \tag{13}$$

**Sampling** $\{\boldsymbol{\pi}_i^{(l)}\}_{i,l}$    After obtaining the latent counts for each layer, the posterior inference in $\boldsymbol{\pi}_i^{(l)}$ can be proceeded as:

$$\boldsymbol{\pi}_i^{(l)} \sim \text{Dirichlet}(\psi_{i1}^{(l)} + m_{i1}^{(l)}, \ldots, \psi_{iK}^{(l)} + m_{iK}^{(l)}) \tag{14}$$

**Sampling** $\{\boldsymbol{B}_{i'i}^{(l)}\}_{i',i,l}$    For $B_{i'i}^{(l)}$, the likelihood can be represented as (following Eq. (7)):

$$\mathcal{L}(B_{i'i}^{(l)}) \propto e^{B_{i'i}^{(l)} \cdot \log q_{i'}^{(l)}} \left(B_{i'i}^{(l)}\right)^{\sum_k h_{i'ik}^{(l)}} \tag{15}$$

For $R_{i'i} \neq 0 \cap i' \neq i$, the prior for $B_{i'i}^{(l)}$ is $\text{Gam}(\gamma_1^{(l)}, \frac{1}{c^{(l)}})$, the posterior distribution is

$$B_{i'i}^{(l)} \sim \text{Gam}(\gamma_1^{(l)} + \sum_k h_{i'ik}^{(l)}, \frac{1}{c^{(l)} - \log q_{i'}^{(l)}}) \tag{16}$$

For $i' = i$, the prior for $B_{ii}^{(l)}$ is $\text{Gam}(\gamma_0^{(l)}, \frac{1}{c^{(l)}})$, the posterior distribution is

$$B_{ii}^{(l)} \sim \text{Gam}(\gamma_0^{(l)} + \sum_k h_{iik}^{(l)}, \frac{1}{c^{(l)} - \log q_i^{(l)}}) \tag{17}$$

**Sampling** $\{X_{ik}\}_{i,k}$**:**    From the Poisson-Multinomial equivalence [1] we have $M_i \sim \text{Poisson}(M)$,

$$(X_{i1}, \ldots, X_{iK}) \sim \text{Multi}(M_i; \pi_{i1}^{(L)}, \ldots, \pi_{iK}^{(L)}) \overset{d}{=} X_{ik} \sim \text{Poisson}(M \pi_{ik}^{(L)}), \forall k.$$

Both the prior distribution for generating $X_{ik}$ and the likelihood parametrised by $X_{ik}$ are Poisson distributions. The full conditional distribution of $X_{ik}$ (assuming $z_{ii,..} = 0, \forall i$) is then

$$P(X_{ik}|M, \boldsymbol{\pi}, \boldsymbol{\Lambda}, \boldsymbol{Z}) \propto \frac{\left[M \pi_{ik}^{(L)} e^{-\sum_{j \neq i, k_2} X_{jk_2}(\Lambda_{kk_2} + \Lambda_{k_2 k})}\right]^{X_{ik}}}{X_{ik}!} (X_{ik})^{\sum_{j_1, k_2} Z_{ij_1, kk_2} + \sum_{j_2, k_1} Z_{j_2 i, k_1 k}}. \tag{18}$$

This follows the form of Touchard polynomials [2], where $1 = \frac{1}{e^x T_n(x)} \sum_{k=0}^{\infty} \frac{x^k k^n}{k!}$ with $T_n(x) = \sum_{k=0}^n \{_k^n\} x^k$ and where $\{_k^n\}$ is the Stirling number of the second kind. A draw from (18) is then available by comparing a Uniform$(0, 1)$ random variable to the cumulative sum of $\{\frac{1}{e^x T_n(x)} \cdot \frac{x^k k^n}{k!}\}_k$.

**Sampling** $\{Z_{ij,k_1 k_2}\}_{i,j,k_1,k_2}$    We first sample $Z_{ij,\cdot\cdot}$ from a Poisson distribution with positive support:

$$Z_{ij,\cdot\cdot} \sim \text{Poisson}_+(\sum_{k_1,k_2} X_{ik_1} X_{jk_2} \Lambda_{k_1 k_2}), \text{where } Z_{ij,\cdot\cdot} = 1,2,3,\ldots \tag{19}$$

Then, $\{Z_{ij,k_1 k_2}\}_{k_1,k_2}$ can be obtained through the Multinomial distribution as:

$$(\{Z_{ij,k_1 k_2}\}_{k_1,k_2}) \sim \text{Multinomial}\left(Z_{ij,\cdot\cdot}; \left\{\frac{X_{ik_1} X_{jk_2} \Lambda_{k_1 k_2}}{\sum_{k_1,k_2} X_{ik_1} X_{jk_2} \Lambda_{k_1 k_2}}\right\}_{k_1,k_2}\right) \tag{20}$$

**Sampling** $\{\Lambda_{k_1 k_2}\}_{k_1,k_2}$    For $\Lambda_{k_1 k_2}$'s posterior distribution, we get

$$P(\Lambda_{k_1 k_2}|-) \propto \exp\left(-\Lambda_{k_1 k_2}(\sum_{i,j} X_{ik_1} X_{jk_2})\right) \Lambda_{k_1 k_2}^{\sum_{i,j} Z_{ij,k_1 k_2}} \cdot \exp\left(-\Lambda_{k_1 k_2}\theta_\Lambda\right)\Lambda^{k_\Lambda - 1} \tag{21}$$

Thus, we get

$$\Lambda_{k_1 k_2} \sim \text{Gam}\left(\sum_{i,j} Z_{ij,k_1 k_2} + k_\Lambda, \frac{1}{\theta_\Lambda + \sum_{i,j} X_{ik_1} X_{jk_2}}\right) \tag{22}$$

**Sampling** $M$    $M_i$'s posterior distribution is:

$$P(M|-) = M^{k_M - 1} \exp(-\theta_M M) \prod_{i,k} \left(\exp(-M\pi_{ik}^{(L)})\right) M^{\sum_{i,k} X_{ik}} \tag{23}$$

Thus, we sample $M$ from:

$$M \sim \text{Gam}\left(k_M + \sum_{i,k} X_{ik}, \frac{1}{\theta_M + N}\right) \tag{24}$$

**Sampling** $\alpha$    Similarly, $\alpha$'s posterior distribution is

$$\alpha \sim \text{Gam}(k_\alpha + \sum_{i,k} h_{i\alpha k}^{(1)}, \frac{1}{\theta_\alpha - \sum_i \log q_i^{(1)}}) \tag{25}$$

**Sampling hyper-parameters of** $\Lambda$    We set the following distributions for the hyper-parameters:

$$k_\Lambda \sim \text{Gam}(k_2, \frac{1}{\theta_2}), \theta_\Lambda \sim \text{Gam}(k_3, \frac{1}{\theta_3}) \tag{26}$$

The posterior distribution of these hyper-parameters are:

$$l_{k_1 k_2} \sim \sum_{t=1}^{\sum_{i,j} Z_{ij,k_1 k_2}} \text{Ber}\left(\frac{k_\Lambda}{k_\Lambda + t - 1}\right), \quad k_\Lambda \sim \text{Gam}(k_2 + \sum_{k_1,k_2} l_{k_1 k_2}, \frac{1}{\theta_2 - \sum_{k_1,k_2} \log(1 - p'_{k_1 k_2})})$$

$$\theta_M \sim \text{Gam}(k_3 + K^2 k_\lambda, \frac{1}{\theta_3 + \sum_{k_1,k_2} \Lambda_{k_1 k_2}}) \tag{27}$$

where $p'_{k_1 k_2} = \frac{\sum_{i,j} X_{ik_1} X_{jk_2}}{\theta_\Lambda + \sum_{i,j} X_{ik_1} X_{jk_2}}$.

**Sampling hyper-parameters of** $\beta$    We set the following distributions for the hyper-parameters:

$$\gamma_1^{(l)}, \gamma_0^{(l)} \sim \text{Gam}(e_0^{(l)}, \frac{1}{f_0^{(l)}}), c^{(l)} \sim \text{Gam}(g_0, \frac{1}{h_0}) \tag{28}$$

The posterior distribution of these hyper-parameters are:

$$J_{i'i}^{(l)} \sim \text{CRT}(\sum_k h_{i'ik}^{(l)}, \gamma_1^{(l)}), \forall(i',i)|R_{i'i} = 1 \cap i' \neq i$$

$$J_{ii}^{(l)} \sim \text{CRT}(\sum_k h_{iik}^{(l)}, \gamma_0^{(l)}), \forall i$$

$$n_1^{(l)} = \sum_{(i,i')|i \neq i' \cap R_{i'i}=1} \log \frac{c^{(l)} - \log q_i^{(l)}}{c^{(l)}}, n_0^{(l)} = \sum_i \log \frac{c^{(l)} - \log q_i^{(l)}}{c^{(l)}}$$

$$\gamma_1^{(l)} \sim \text{Gam}(e_0 + \sum_{i \neq i'} J_{i'i}^{(l)}, \frac{1}{f_0 + n_1^{(l)}})$$

$$\gamma_0^{(l)} \sim \text{Gam}(e_0 + \sum_{i'} J_{i'i'}^{(l)}, \frac{1}{f_0 + n_0^{(l)}})$$

$$c^{(l)} \sim \text{Gam}(g_0 + N\gamma_0^{(l)} + \gamma_1^{(l)} \sum_{i \neq i'} \mathbf{1}(R_{ii'} = 1), \frac{1}{h_0 + \sum_{i,i'} \beta_{i'i}^{(l)}}) \tag{29}$$

---

**Algorithm 1** Sampling for SDREM
---
**Input:** relational data $\{R_{ij}\}_{i,j=1}^N$, nodes' feature information $\boldsymbol{F} \in (\mathbb{R}^+ \cup 0)^{N \times D}$, iteration time $T$
**Output:** $\{\boldsymbol{\pi}_i^{(l)}\}_{i,l}, \{\boldsymbol{B}^{(l)}\}_{l=1}^{L-1}, \{\boldsymbol{X}_i\}_i, \{\Lambda_{k_1 k_2}\}_{k_1, k_2}, \boldsymbol{T}, \alpha, M$

  **for** $t = 1, \ldots, T$ **do**
    // Update the latent counts in a bottom-up manner
    **for** $l = L, \ldots, 2$ **do**
      Update latent count vector $y_{ik}^{(l)} \sim \text{CRT}(m_{ik}^{(l)}, \psi_{ik}^{(l)})$
      Update the latent count on the $l$-layer Eq. (6)
      Summarize the input $m_{ik}^{(l-1)}$ for $(l-1)$-th layer Eq. (9), $\forall i, k$
    **end for**
    Update latent count vector $y_{ik}^{(1)} \sim \text{CRT}(m_{ik}^{(1)}, \psi_{ik}^{(1)})$
    Update the latent count on the 1st-layer Eq. (11)
    // Update $\{\boldsymbol{\pi}_i^{(l)}\}_{i,l}$ and $\{B_{i'i}^{(l)}\}_{i,l}$ from the input layer to the output layer
    Update $\{T_{dk}\}_{d,k}$ according to Eq. (13)
    **for** $l = 1, \ldots, L$ **do**
      Update membership distribution $\boldsymbol{\pi}_i^{(l)}$ Eq. (14)
    **end for**
    **for** $l = 2, \ldots, L$ **do**
      Update coefficients $B_{i'i}^{(l)}$ according to Eq. (16)(17), $\forall i', i$
    **end for**
    // Update relational data generation structure
    **for** $l = i, \ldots, N, k = 1, \cdots, K$ **do**
      Update latent counts $X_{ik}$ according to Eq. (18)
    **end for**
    **for** $(i, j)|R_{ij} = 1$ **do**
      Update latent representation $\{Z_{ij,k_1 k_2}\}$ according to Eq. (19)(20)
    **end for**
    **for** $k_1, k_2 = 1, \ldots, K$ **do**
      Update compatibility value $\{\Lambda_{k_1 k_2}\}$ according to Eq. (22)
    **end for**
    Update $\alpha, M$ according to Eq. (25)(24)
    Update hyper-parameters of $\boldsymbol{\Lambda}, \boldsymbol{\beta}$ according to Eq. (27),(29)
  **end for**
---

Figure 1: Left: visualizations on the membership distributions ($\{\boldsymbol{\pi}_{1:50}^{(l)}\}_{l=1}^3$) and normalized auxiliary counting variable ($\bar{\boldsymbol{X}}_{1:50}$) for the first 50 nodes of the *Cora, PPI, Pubmed* datasets (row represents the nodes and column represents the latent features); right: visualizations on the non-zero positions ($\boldsymbol{R} + \boldsymbol{I}$) and transition coefficient matrix ($\{\boldsymbol{\beta}^{(l)}\}_{l=1}^2$) for the first 200 nodes of the *Cora, PPI, Pubmed* datasets.

Figure 2: Compatibility matrix for the datasets of *Citeer, Cora, PPI, Pubmed*.

## 2 Latent feature visualization for the datasets of Citeer, Pubmed and PPI

We provide the visualizations on latent features for the datasets of Citeer, Pubmed and PPI in Figure 1. Similar conclusions (as mentioned in the main paper) can be obtained.

We also provide visualization on the compatibility matrix $\boldsymbol{\Lambda}$.