[Reviews · NeurIPS 2019]

Reviewer 1



1). First of all, I don't think it's a good way to represent each node with a dirichlet distribution leading to a positive node embedding. It's quite different from traditional real-valued embedding methods and I assume positive embedding representations will directly reduce semantic information compared to real-valued. So if there are any other positive embedding methods, please refer them to illustrate the relation to the proposed method. 2) . As mentioned in the article, the proposed SDREM propagating information through neighbors works in a similar spirit to the spatial graph convolutional network (GCN) in a frequentist setting. But as far as I am concerned, GCNs that have already been applied, will not only consider neighboring information in graphs, but also propagate each node embedding to a deeper representation through a fully connected network. Different from traditional GCNs, the proposed SDREM only summarizes the entire neighboring node embeddings with the learned weight, so please provide a conceptual comparison between the proposed SDREM and GCNs as described in [1]. 3) . Similar in experimental parts, the authors give a comparison with GCN as described in [2], but why not compare with the improved version in [1]? 4) . There are some other related work [3], which has a similar structure with SDREM, please give a comparison on model aspect and it will be better if you can list quantitative comparisons in experiments. 5) More test details should be shown. The relationship between nodes is needed during the training process. I don't understand how to do the test. 6) The latent representation pi is interpretable as node i’s community distribution, however, the number of communities K in each layer keeps the same in SDREM. Could the authors give the reasons? 7) why not model R_ij as a Bernoulli distribution ? This is more in line with the Bayesian model. After all, the authors introduced a Bayesian framework by using deep latent representations for nodes to model relational data. And it's a quite novel model although there are many places to improve. [1]. Kipf, Thomas, and Max Welling. Semi-Supervised Classification with Graph Convolutional Networks[J]. In ICLR, 2017. [2]. Diederik P. Kingma, Danilo Jimenez Rezende, Shakir Mohamed, and Max Welling. Semisupervised learning with deep generative models. In ICLR, 2014. [3]. Hu, Changwei, Piyush Rai, and Lawrence Carin. Deep Generative Models for Relational Data with Side Information. In ICML, 2017

Reviewer 2



Originality: The main originality of the paper is the proposal of a novel model construction for Bayesian graph models. Although related hierarchical constructions have been adopted in other areas like topic modelling, the proposed model is relatively new in Bayesian modelling for graph analysis. Quality: The quality of the paper is good, especially supported by comprehensive experiments. Specifically, the datasets and baseline methods are well chosen and detailed experiments are provided for ablation study and different model settings. Those experiments show the claims of the paper. However, there can be several improvements to the paper: 1. Can the authors provide some quantitative study of the proposed model? It is known that one of the advantages of Bayesian models over neural networks is interpretability. It would be better if the authors can show some quantitative analysis for intuitively understanding. 2. According to my experience, AUC-PR (Precision-Recall) can be a better measurement over AUC-ROC (reported in this paper) when the proportion of training data is low. Therefore, it would be better to see that AUC-PR is also reported in the revised version of the paper in the experiment of Figure 4. Clarity: The paper is well-written as easy to follow. My comments on clarity are as follows: 1. Can the authors further explain why it is necessary to draw X_{i1} ... X_{iK} from multinomial (Figure 1 (3) )? As far as I can see, one can directly feed \pi_{i1} ... \pi{1K} into Figure 1 (5). I am not sure the intuition on the drawing of X. 2. Page 5 Line 183, Eq. (18) does not exist in the main paper. Significance: Link prediction is a very important task in graph analysis. My major concern of the proposed model is on the scalability of Gibbs sampling, which may prevent the model to be applied in larger graphs. Although the authors claim that the model is scalable, the conducted experiments are mainly on moderately small datasets. ----------------------------------------------------------------------------------------- After the author response: Thanks for the response. I think it's an interesting paper. But several concerns were raised by the other reviewers and me, which may need further improvements, including more comprehensive comparisons with GCN and VGCN, and qualitative analysis on interpretation of the multi-layer structure. After the discussion, I changed my original score.

Reviewer 3



The paper presents a deep generative model for relational data. There is a considerable recent interest in modeling graoh-structured data, not only in traditional domains, such as social network analysis, but also in domains such as molecular design. Therefore, it is a timely contribution. As is common witrh deep architectures, the paper assumes that each node in the graph has a deep representation with each layer's representation depending on the above layer's representation of that node (and other "adjacent" nodes), which enables capturing higher order node dependence. The representation in each layer is modeled by a Dirichlet random variable, which in turn is conditioned on node features (if available). Comments: - The idea seems to be inspired by recent work (Hu et al, 2017) and properties of Bernoulli-Poisson link function. However, the paper also makes some key improvements over previous work (e.g., higher order node dependencies), which results in better predictive accuracies. - Inference for the model is done via MCMC sampling which for the proposed model is expensive and fairly complicated (relies on latent variable augmentation techniques that facilitate closed form sampling updates). This somewhat limits the scalabilty to really massive datasets. Although the cost scales in the number of edges in the graph, the method also requires sampling a large number of other latent variables which could potentially make it somewhat expensive for large graphs. - The paper should also discuss other recent work on deep generative models for graphs, such as (1) Variational Graph Autoencoder (Kipf and Welling, 2016) (2) Stochastic Blockmodels meet Graph Neural Networks (Mehta et al, 2019) These methods use graph convolutional network (GCN) based encoder to learn latent representation in a generative model (latent space model or stochastic blockmodel). It would also be nice to have a discussion /comparison with such models. - For the experiments, since the proposed methods can use node features, it might be a good idea to evaluate it on "cold-start" problems, i.e., hold out all the link/non-links for some nodes and try to see if the model can predict them well. - The model is fairly complicated and MCMC is used for inference. In the experiments, 2000 iterations were run. Were these sufficient? Did the MCMC chain seem to converge? If yes, how did you assess that? - The model is fairly involved (though inspired by previous works) and even though the inference update equations are provided, implementing it from scratch would be non-trivial. Moreover, the experimental settings, hyperparameter etc. need to be carefully selected. It would be nice if the code can be made available (I could not find it with the submission) for easy reproducibility Overall, the paper is addressing an important problem. The proposed model, although inspired by various recent works, makes some interesting improvements. The empirical evaluation is decent; however, it is limited to small-sized graphs. ========================================= Post rebuttal comments: Thanks for your response. I still feel the paper, despite having an interesting model and decent experiments, is somewhat borderline, especially since the basic model is largely inspired by HLFM [11] and the MCMC based inference method used in the paper is mostly based on prior work. Also, although the model is interesting, the paper falls short on making a strong case as to why it would be preferable as compared to various other deep generative models of graphs proposed in the recent couple of years. I am especially talking about generative models that use things like graph convolutional networks (GCN) and recurrent neural networks as encoder. The paper doesn't discuss or compare against any of these methods, which is disappointing since these methods are now very widely used in modeling of graph structured data. Perhaps one of the most basic of these is the VGAE (variational graph autoencoder) or Kipf and Welling (2016) which should have been a simple baseline. VGAE uses GCN and does incorporate higher order node correlations due to the message-passing scheme of GCN, and it would be nice to show a comparison with such methods. Even if the paper doesn't compare with more advanced variants of VGAE, at least some basic comparison with VGAE should have been provided. The proposed model does have some nice interpretability properties but the paper doesn't explore those. A nice experiment would be to use the embeddings to discover topics in the graph (for example, as done in HLFM [11]). In summary, the paper has promising ideas but, in my opinion, it is somewhat found lacking due to the other reasons. I think by incorporating the above suggestions, the paper would be significantly better. As I said, the paper is borderline. If the paper does get accepted, I would encourage the authors to include some comparisons with VGAE based methods (and a discussion of pros/cons with Kipf and Welling (2016) and Mehta et al (2019)). Some qualitative results would also be nice to have.

[Author Response · NeurIPS 2019]

# Paper ID 6878: Response to Reviewers

Firstly, a big thankyou to all Reviewers for their time and constructive comments. We are able to address ALL raised queries and concerns (detailed below). Any points not directly addressed below will be corrected as a matter of course.

**For Reviewers 2 & 3 on the presented data augmentation trick**: As the prior for node embedding $\boldsymbol{\pi}_i^{(L)}$ is not a conjugate prior for the Ber-Poisson likelihood in SDREM, we can not obtain a Gibbs sampling update for $\boldsymbol{\pi}_i^{(L)}$. To circumvent this, we introduce an auxiliary latent counting vector $\boldsymbol{X}_i$. On one hand, each element $X_{ik}$ in $\boldsymbol{X}_i$ is generated by $\pi_{ik}$ only; on the other hand, the whole vector $\boldsymbol{X}_i$ can be regarded as a draw from a Multinomial distribution (with $\boldsymbol{\pi}_i$ as event probabilities). In this way, Gibbs sampling is then permitted for the Ber-Poisson likelihood function in this setting. We will clarify this novel construction in the revised version.

**For Reviewers 2 & 3 on scalability**: As SDREM uses Gibbs sampling, it does have some limitations in scaling to large networks. However the computational cost is only $\propto$ the number of positive links. We will clarify in the text.

## Response to Reviewer 1

Reviewer 1 queries the lack of comparison with two specific and relevant methods. However, each requested comparison method (i.e. the GCN of reference [1] and the HLFM of reference [3], in Reviewer 1's comments) **has already been executed and compared** in the paper. Please see our responses for **2)–4)** below to explain this.

**1) Referring to other positive embedding methods**: As mentioned (Lines 208–209, 176–177) our positive-valued embedding method is related to the previous positive-valued embedding methods: Gamma Belief Networks [30] and Dirichlet Belief Networks [28]. We will improve the clarity of the text in the revised version.

**2) and 3) Comparison with GCN**: The GCN algorithm we discussed (Lines 127–134) and quantitatively compared against (Fig. 4) is actually Reviewer 1's requested GCN algorithm! We accidentally used the wrong reference in the text when citing the GCN algorithm. Thank you for your considered comment that allows us to spot and correct this.

**4) Comparison with HLFM**: Our reference [11] (HLFM) is reference [3] in Reviewer 1's comments. So we have already compared our SDREM with the HLFM in the 3rd paragraph in Section 4 (Lines 199-204) and in Fig. 4.

**5) Test details**: The testing relational data are not used when constructing the information propagation matrix (i.e. we set $\beta_{i'i}^{(l)} = 0$ if $R_{i'i}$ is testing data). We will clarify this in the revised version.

**6) Same size of node embedding**: For modelling simplicity, we used equal sizes for the node embeddings. However, by using merge-and-split (Beta distributed splitting ratio) operations on the elements of the Dirichlet distributions, the node embeddings can be of different sizes while still permitting Gibbs sampling on all variables. We will clarify this.

**7) Why does $R_{ij}$ not follow Bernoulli distribution?** Actually, $R_{ij}$ DOES follow the Bernoulli distribution when we integrate out the latent integers $Z$ (see Lines 136-137). We will improve the text on this point.

## Response to Reviewer 2

**Analysis for intuitive understanding**: Thankyou for the suggestion. We will provide some interpretable visualisations on some embedding outcomes in the revised version and comment that this is an advantage of the Bayesian model.

**Reporting AUC-PR (Precision-Recall) values**: This is a fair point (thank you). We have now calculated AUC-PR values for our analyses, and the results are consistent with our previous conclusions. We will update our results and discussion to incorporate this additional qualitative assessment.

## Response to Reviewer 3

**Comparisons with graph VAE methods (VGAE)**: As Mehta et al (2019) was available on arXiv ONLY 9 days before the NeurIPS submission deadline, we missed it when submitting SDREM. VGAE has a larger computational complexity ($\mathcal{O}(N^2)$). It uses parameterized functions to construct the deep network architecture and the probabilistic nature occurs in the output layer as Gaussian random variables only. In contrast SDREM constructs fully probabilistic deep architectures (with Dirichlet random variables at each layer). We will highlight these differences in the revision.

**Evaluating SDREM on "cold-start" problems**: Thankyou for the idea. We ran a quick experiment (following the recommended settings and with train:test = 9:1). The average AUC values are: Citeer (0.653), Cora (0.667), PPI (0.837), Pubmed (0.761), showing the effectiveness of our SDREM in using feature information. We will update the paper.

**Diagnosing the convergence of MCMC algorithms**: Actually 2000 iterations were adequate here (mixing was good). We used visual trace plots and standard convergence tests (on AUC & negative log-likelihood) to assess convergence.

**Open source code**: Yes, we will of course open source the code once the paper is published.

[Meta-Review · NeurIPS 2019]

The paper was reviewed by three experts in the field. The reviewers and AC all agree that the paper contains novel contributions, but share the same opinion that it could be strengthened by addressing the reviewers' comments. In addition to the reviewers' comments such as the need to adding comparison with VGAE and its variates, the AC would like to provide some additional feedback to the authors: The AC views the paper as some kind of smart combination of edge partition model, gamma belief net, and Dirichlet belief net, enhanced by adding covariate dependence and by incorporate the network information in learning the connection weights of the Dirichlet belief net. Pros: 1) the combination is non-trival: replacing the gamma weights in edge partition model with latent counts is the key to allow closed-form Gibbs sampling (upward latent count propagation followed by downward variable sampling). How the X is used in (3) and sampled in (5) is novel. 2) the way the network information is used to build the Dirichlet belief net, as shown in (2), is smart and seems novel. 3) showing an interesting connection that the proposed model is related to mixed-membership stochastic blockmodel, but does not suffer from the same issue of having O(N^2) computation. 4) the authors have mastered a number of non-trival data augmentation techniques for multinomial/Poisson/gamma/Dirichlet variables and nicely combined them to derive Closed-form Gibbs sampling updates. Both Good and Bad: While improving Edge partition model with node covariates has been considered before by HLFM [11], where the gamma weights have been replaced with binary weights, here the authors use count-valued weights modeled with a Dirichlet belief net whose N*N connection weights are sparsified by the N*N network adjacency matrix. Cons: 1) The adjacency matrix, which the hierarchical model is trying to model, has been used to parameterize the Dirichlet belief net prior, which makes the model no-longer an exact generative model. 2) The comparison with HGP-EPM in Figure 4 might not be that fair, as HGP-EPM is not designed to include covariates (the authors are aware of that, as in Line 286-287). It is unclear to me why the Plain-SDREM (i.e., SDREM without node covariates) is not used for comparison with HGP-EPM. 3) In addition to having close-form Gibbs sampling updates, a great defense for the authors to argue against the need to compare with GCN and VGAE is demonstrating the interpretability of the proposed model, e.g., showing communities memberships and overlapping community structures, as it has been done in both EPM [29] and HLFM [11]. 4) It could follow HGP-EPM to model the Lambda matrix with a relational hierarchical gamma process prior, and hence could potentially infer the number of communities. Unfortunately, the authors choose to impose an i.i.d. gamma prior instead and hence could no longer infer K. Related to this point, Figure 2 second panel shall further increase K, which may show performance drop when K is over certain size (which could be prevented if the relational hierarchical gamma process prior is used). In summary, the paper does have clear novel contributions, but also has clear room for improvement. The authors are encouraged to carefully revise their paper to further enhance its quality.